# The Evaluation of a Mindful Coaching Programme to Reduce Burnout in Social Workers in Hong Kong—A Pilot Study

**DOI:** 10.3390/bs14100915

**Published:** 2024-10-08

**Authors:** Camille K. Y. Chan, Paul W. C. Wong

**Affiliations:** Department of Social Science and Social Administration, The University of Hong Kong, Hong Kong SAR, China; cchanky@connect.hku.hk

**Keywords:** burnout, depression, compassion fatigue, compassion satisfaction, ProQOL, burnout, secondary traumatic stress, healthy boundaries, social workers, helping professionals

## Abstract

Social work professionals experience high levels of burnout, emotional exhaustion, and secondary traumatic stress (STS). This paper reports the findings of a pilot test of the Burn-Not-Out programme that aimed to reduce social workers’ burnout, STS, and depressive symptoms and to enhance compassion satisfaction (CS), using the mixed methods evaluation methodology. Hong Kong social workers participated in the programme between December 2023 and March 2024 and completed pre- and post-programme self-administered online surveys including the Professional Quality of Life (ProQOL) and Patient Health Questionnaire (PHQ-9), healthy alongside self-constructed questions on boundary setting, and post-programme online focus group discussions. The results from 94 paired surveys revealed a concerning mental health profile of the participants at baseline, and, after programme participation, there were statistically significant reductions in burnout (Cohen’s d = 0.73) and depressive symptoms (Cohen’s d = 0.57) among the participants. The participants in the focus group reported that they valued the programme’s emphasis on healthy boundaries, one-on-one coaching, and the sense of being cared for, which contributed to their mental health improvement. This study highlights the urgent need for more research on the role of psychological capital in social workers’ resilience and calls for more empirical systemic interventions that can promote social workers’ mental wellness, with sustainable policies that ensure manageable workloads and adequate workplace support.

## 1. Introduction

Social work is a demanding profession, requiring practitioners to support vulnerable populations with complex needs. Being at the forefront of clients’ personal and family crises often results in being exposed to high levels of stress. Burnout, characterised by emotional exhaustion, depersonalisation, and reduced personal achievements, is closely linked to mental health issues such as anxiety, depression, and somatic symptoms, as well as physical health deterioration [1,2,3]. A meta-analysis revealed that social workers in social services experience a 20% prevalence of burnout, with emotional exhaustion (50%), depersonalisation (45%), and a lack of personal accomplishment (39%) being the primary contributors [4]. Moreover, secondary traumatic stress (STS) from clients’ trauma exacerbates emotional exhaustion, burnout, and psychological distress [5,6,7].

Compassion fatigue, encompassing burnout and STS, intensifies this strain on social workers [5,8], contributing to elevated rates of depression and anxiety, with depression affecting between 19% and 51% of social workers [9,10]. Depression is a significant predictor of compassion fatigue [11,12] and can be further complicated by contextual factors, such as administrative responsibilities under unsettled societal and political environments [7]. For instance, dissatisfaction with income, experience of violence, feeling undervalued, excessive job demands, and a limited decision-making latitude compound these stressors [9,13,14], negatively impacting both social workers’ well-being and the quality of care they provide [6].

Protective factors, especially compassion, against burnout in helping professionals is well documented. Practicing self-compassion, a form of compassion directed toward oneself, has been consistently linked to reductions in both burnout and depression [15,16]. Similarly, individuals who engage in stress reduction techniques report higher levels of compassion satisfaction (CS) [17], which is the sense of fulfilment from helping others through compassionate work. Mindfulness practices, which promote emotional awareness and stress management, have been shown to reduce compassion fatigue and enhance well-being [18,19]. Setting clear boundaries in work relationships further mitigates emotional strain, reducing burnout risk [20]. Supportive work environments, job autonomy, and manageable workloads are also crucial for fostering professionals’ resilience [7,21,22]. Moreover, psychological capital [23], self-care [24], finding intrinsic value in one’s profession [21], and having a strong sense of purpose in life [25] can also serve as protective measures against burnout and emotional exhaustion.

However, despite these insights, Hong Kong social workers often lack access to the necessary emotional support and resources to counter burnout. A previous study revealed that over 90% of social workers in Hong Kong considered themselves overloaded with work [26], further compounding the city’s broader mental health crisis, where 73.7% of surveyed citizens have shown moderate-to-severe symptoms of depression [27], and there has been anecdotal evidence suggesting that the situation has worsened in the post-COVID period. According to the Social Worker Registration Board, there are 28,622 registered social workers serving about 7.4 million people in Hong Kong [28]. While burnout is widely recognised as a critical concern, there is currently a lack of specific interventions and limited research on the extent and impact of burnout in Hong Kong social workers.

This research evaluated the “Social Workers Burn-Not-Out programme” (hereafter, Burn-Not-Out) developed by Dreams Possible, a Hong Kong-based charity, which was an eight-session hybrid mindful coaching programme aiming to reduce burnout among Hong Kong social workers. This study examined its effectiveness before and after the programme, alongside participants’ experiences with burnout to evaluate the programme’s effectiveness in addressing it. We hypothesised that Burn-Not-Out participants would show reduced levels of burnout, STS, and depressive symptoms; increased CS; and an improved ability to draw healthy boundaries after the programme.

## 2. Materials and Methods

### 2.1. The Programme

Burn-Not-Out, developed by Dreams Possible, was a holistic mindful coaching programme specifically developed for social workers in Hong Kong that aimed to reduce their burnout levels through mindset transformation, self-healing promotion, and empowerment. It focused on five core components: (i) grounding for experiential learning, (ii) self-coaching tools to cultivate self-compassion and self-esteem, (iii) individualised one-on-one coaching, (iv) communal support beyond the programme through WhatsApp, and (v) a 3 min daily practice to ensure long-term sustainability.

The 30 h programme comprised eight sessions and was divided into three stages: (1) ‘being coached’, which promoted self-healing and purpose discovery; (2) ‘self-coaching’, to reshape identity; and (3) ‘coaching others’, which taught skills for deep connection and complex case management. The sessions, led by qualified coaches, incorporated the 3A model (acknowledge, act, appreciate), inner child transformation, and boundary setting. This hybrid programme took place between 2 January and 1 March 2024, blended with lectures, one-on-one coaching, group discussions, and practical demonstrations.

### 2.2. Study Design

The Burn-Not-Out programme was pilot-tested and evaluated using a mixed method approach. Quantitative data were collected through self-administered online surveys before and after the programme. The pre-programme survey captured the baseline levels of burnout, STS, CS, healthy boundaries, and depressive symptoms. The post-programme survey, using the same measurements, assessed changes in these outcomes with the following hypotheses:

**Hypothesis 1.** 
*The participants will show reduced levels of burnout, STS, and depressive symptoms after the programme;*


**Hypothesis 2.** 
*The participants will experience increased CS and an improved ability to draw healthy boundaries after the programme;*


**Hypothesis 3.** 
*Burnout levels will be positively associated with depressive symptoms.*


An a priori sample size calculation was performed using G*Power [29]. Assuming a medium effect size (Cohen’s d = 0.5), with a power of 0.8 and a significance level of 0.05 (two-tailed), a minimum of 34 participants were required for this study’s paired *t*-test analysis.

The qualitative component of this study was grounded in a post-positivist epistemological perspective, recognising that, while objective reality exists, our understanding of it is inevitably shaped by individual experiences and contexts [30]. The focus group discussions aimed to explore the participants’ lived experience with burnout, assess the effectiveness of the programme, and identify potential areas for refinement in future iterations. This study intended to conduct two semi-structured focus group discussions, with each lasting for approximately 60 min, targeting six to eight participants, allowing for in-depth discussion and varied perspectives [31].

### 2.3. Setting and Participants

All the participants enrolled in the Burn-Not-Out programme were invited to participate in the quantitative study, involving self-administered online surveys before the programme’s commencement and after its final session. Anyone who was enrolled in the programme was eligible for this study. The interested participants were guided to an online survey platform [32], where they were provided with an information sheet and consent form in traditional Chinese to ensure informed consent. After consenting, the participants completed e-screening questions and, if eligible, proceeded to the self-administered survey. The same procedure was followed for both the pre- and post-programme surveys. The pre-programme data collection occurred between 27 December 2023 and 8 January 2024, while the post-programme data were collected between 26 February and 11 March 2024.

For the qualitative phase, the participants who completed the programme were eligible to join the focus group discussions. The online consent form was distributed via WhatsApp, and, subsequently, the online link for the focus group discussion was also shared in WhatsApp, once we obtained their consent. The individuals who did not consent were excluded from this study. The two focus group discussions were conducted on 12 and 13 March 2024.

The recruitment for both phases was facilitated by Dreams Possible, who used the WhatsApp group and in-class announcements to promote participation. The research team maintained strict confidentiality, ensuring the participants’ anonymity by not disclosing to anyone, including Dreams Possible, who participated in the surveys or focus group discussions.

### 2.4. Measurements

The quantitative online surveys incorporated validated tools, including the Professional Quality of Life scale (ProQOL) [8] and the Patient Health Questionnaire-9 [33], as well as self-constructed questions on healthy boundaries. The participants’ demographics and work-related backgrounds were also gathered, including their age, gender, experience, job roles, and work hours.

#### 2.4.1. Professional Quality of Life Scale (ProQOL)

ProQOL is an internationally validated scale that measures the compassion fatigue and CS of helping professionals on a 5-point Likert scale. The 30-item scale included subscales for burnout, STS, and CS, with scores categorised as low (22 or below), moderate (23 to 41), or high (42 or above) [8]. The Cronbach α value of each subscale was tested with the Hong Kong helping professionals, with values ranging from 0.79 to 0.93 [12], indicating a good internal consistency and reliability.

#### 2.4.2. Patient Health Questionnaire-9 (PHQ-9)

This validated 9-item scale assesses depressive symptoms and current suicidal ideation on a scale from 0 (not at all) to 3 (nearly every day) [33]. The summed scores, ranging from 0 to 27, can be categorised into five categories from none to severe depressive symptoms. Its excellent internal reliability, with a Cronbach α value of 0.86 [12], was tested with the Hong Kong helping professionals.

#### 2.4.3. Healthy Boundaries

Dreams Possible observed that a social worker’s lack of healthy boundaries with their clients could lead to the overwhelming experience of handling their clients’ emotions and responsibilities, blurring the line between the professional’s role and the clients’ own responsibilities. Their experience also suggested that social workers who can decline unreasonable work demands are less prone to burnout. This information guided the construct of three questions relating to healthy boundaries to assess participants’ level of agreement with three statements: “I’m able to confidently reject unreasonable requests at work”, “I can clearly differentiate my work responsibilities versus those that are not mine”, and “I am not burdened by the emotions and responsibilities of my clients”. The statements underwent reliability analysis, resulting in a Cronbach’s α value of 0.77, indicating an acceptable internal consistency.

#### 2.4.4. Semi-Structured Focus Group Discussion

Using a semi-structured discussion guide, the focus group participants were encouraged to reflect on three main areas: (a) the sources of stress specific to Hong Kong social workers and how these challenges affect their professional roles, (b) feedback on the programme, and (c) suggestions for future iterations to better address burnout prevention in the social worker profession. Both focus groups were moderated only by the authors of this manuscript, who facilitated a dynamic, conversational atmosphere to encourage the organic sharing of insights rather than following a strict question and answer structure. The discussions were audio-recorded and later transcribed verbatim for the content analysis.

### 2.5. Data Analysis

For the pre- and post-surveys, descriptive statistics, including means, percentages, and standard deviations, were determined to summarise the participants’ demographic and work-related characteristics. Each ProQOL subscale and PHQ-9 were tested for data normality and homoscedasticity. Correlation and Kruskal–Wallis H tests were performed to evaluate the relationships between the work-related background and outcome variables. A paired *t*-test was conducted to assess the changes in these variables pre- and post-programme. Additionally, there was a missing value analysis to identify any patterns of missing data and potential systematic biases. The statistical analyses were conducted using SPSS for Windows v. 28 [34]. All the significant tests were two-tailed, and the findings with a *p*-value of 0.05 were considered statistically significant.

The focus group discussions were analysed using content analysis to identify the main themes that address the three research questions. The steps involved data preparation, coding scheme development, coding, code revisions, theme and pattern identification, and interpretation [35]. The audio records were transcribed verbatim by the first author. The code frame was developed deductively with references to the ProQOL [8] and RE-AIM [36] frameworks, as well as inductively from the data collected. The accuracy and completeness of the research findings were validated through discussions among the research team members. Qualitative analysis software aided the coding and theme formation process [37].

The study protocol was reviewed and approved by the Human Research Ethics Committee at the University of Hong Kong (reference number: EA230588).

## 3. Results

### 3.1. Quantitative Results

A total of 255 responses from the pre-programme and 113 from the post-programme surveys were collected and analysed. An analysis of the missing data revealed a pattern where participants frequently exited the survey after completing only the demographic section or failing to provide matching ID, though no systematic bias related to the missing data was identified. This resulted in 48.9% of missing data, preventing us from pairing the pre- and post-surveys. Consequently, we discarded the unmatched data and included only the 94 pairable data in the analysis.

#### 3.1.1. Participants’ Demographics and Work-Related Backgrounds

The descriptive analysis and correlation were performed on the 94 sets of paired data The participants’ ages ranged from 23 to 63 years, with a majority being female (84.0%). Over 90% of respondents were full-time social workers, primarily engaged in child and youth services (29.8%) and elderly services (22.3%). The years of experience varied from less than one year to 40 years, with participants working an average of 44 h per week at the time of the survey. Detailed demographic information is presented in Table 1.

The Pearson and Spearman’s rank correlations were conducted to examine the relationships between the variables, depending on the normality assumptions. The Pearson correlations were used for age, years of work, burnout, and STS, while the Spearman’s rank was used for CS and depressive symptoms.

The pre-programme analysis revealed a weak but significant positive correlation between age and CS. Years of work was positively associated with CS and moderately negatively correlated to burnout. Among the outcome variables, burnout showed a positive corelation with both STS and depressive symptoms and was negatively associated with CS. Additionally, STS was moderately correlated with depressive symptoms and weakly negatively correlated with CS, while CS had a weak negative relationship with depressive symptoms (Table 2).

Kruskal–Wallis H tests were performed to assess the association between the participants’ capacity to draw healthy boundaries with burnout, STS, CS, and depressive symptoms. The majority of outcomes at the pre-programme stage were significantly associated with all three statements on boundary setting. However, at the post-programme stage, the third statement did not show any association with these outcomes (Table 3).

#### 3.1.2. Paired Data Analysis

While the data collected in ProQOL’s burnout and secondary traumatic stress subscales followed a normal distribution, its CS subscale and the PHQ-9’s depressive symptoms underwent a Box–Coc transformation to normalise their distribution before proceeding with the paired *t*-tests [38]. The Cohen’s d effect size was calculated for each outcome (Table 4a,b), and it was determined that there is a medium effect of the programme on burnout, STS, CS, depressive symptoms, and drawing healthy boundaries.

The analysis revealed significant improvements in the reduction in burnout, secondary traumatic stress, and depressive symptoms, as well as higher levels of compassion satisfaction and being better at drawing boundaries (Table 4).

### 3.2. Qualitative Findings

The two focus group discussions were conducted on 12 and 13 March 2024, gathering data from eleven participants of the Burn-Not-Out programme. The first group had six participants and the second had five, involving nine females and two males. The characteristics of the participants are described in Table 5.

A total of 111 codes were identified, with eight themes emerging. Five themes were related to the stressors of the social work profession, while the other three were related to the Burn-Not-Out programme (Table 6). The following subsections begin with an analysis of the participants’ perception of the stress factors that impact social workers in Hong Kong, which is followed by their feedback on the programme, as well as recommendations for improving its future iterations to better address social workers’ burnout.

#### 3.2.1. Social Workers’ Sources of Stress

Our findings revealed that burnout is a significant contributor to social workers’ stress in Hong Kong, particularly relating to the pressure to fulfil organisational requirements and social workers’ role as their clients’ caregiver. All participants from the focus groups agreed that they felt compelled to prioritise quantity over quality in their work due to an emphasis on targets, particularly in securing funding. This focus on “numbers-chasing”, meaning to get as many service users as possible, leads to feelings being misaligned and detracts from the core values of social work, leaving social workers feeling conflicted about their professional identity and purpose. One participant shared, “*what I find fatal is when we feel like we are just chasing numbers instead of providing actual social work services*”.

A heavy workload, administrative burden, policies, and politics were also identified as sources of stress. Almost all the participants reported facing immense pressure to meet deadlines and fulfil administrative requirements, often compromising their well-being and the quality of services provided. One participant noted, “*I prefer using that time consulting more clients (instead of administrative tasks)*”. Some participants expressed doubts about the relevance of these non-social work tasks and questioned their meaning. A participant expressed their belief that the emphasis of their work had shifted away from the quality of service and explained, “*we never assess the elderly’s experiences after the event… we don’t know if we are catering to their needs*”.

The participants highlighted the unique challenges inherent in social work, particularly the tension between professional responsibility and personal boundaries. Several participants described feeling overwhelmed by client dependency, as one explained, “*I am a school-based social worker … Some parents treat me as their main source of support… Parents who exhibit suicidal tendencies often keeps me up at night, worrying they are going to call me in the middle of the night*”. Another participant expressed their concerns about client safety, explaining that “*you are cautious about strictly following protocols, but you also wouldn’t know if your client can stay safe from their abusive parents*”.

In terms of workplace dynamics, the participants with more work experience mentioned the stress of managing both seniors and subordinates within the social work hierarchy. One participant stated, “*one problem is many of those in managerial positions are not social workers by profession… their mindset can undermine our professional role (despite their good intentions)… For instance, our goal as social workers is to empower our clients, but they want us to spoon-feed them, which potentially foster dependency*”. Inadequate support was also mentioned, which further exacerbates these challenges, with one participant remarking: “*during my time as a school social worker, I only get to see my supervisor once every few months and it was unreasonable*”.

The participants recognised understaffing as a significant recent stressor, fearing that the rapid turnover of experienced professionals was creating a detrimental cycle that added stress to the remaining workforce and was potentially jeopardising the standard of care provided to individuals in need. A few participants agreed that such high turnover rates could lead to feelings of incompetence, negatively affect morale, and lead to burnout. While some emphasised the importance of having harmonious relationships with colleagues to alleviate burnout, they also suggested that these might only serve as temporary reliefs of emotion, with one participant stating that “*these approaches don’t seem to offer practical solutions for the issues at hand*”.

Despite these stressors, the participants recognised that the effort they invest in this compassionate work is meaningful and worthwhile. One participant emphasised, “*supporting youth with suicidal tendencies can be exhausting… but the sense of fulfilment (accompanying them in this difficult journey) is beyond comparison to money and time*”. However, a few participants also revealed that this feeling is often short-lived as they often get overshadowed by the demanding realities, stating that “*because you still have pile of work to do and the organisation or the policies aren’t helping*”.

#### 3.2.2. Participants’ Opinion of the Burn-Not-Out Programme

The participants appreciated the focus on mental wellness in the Burn-Not-Out programme, emphasising how it created a supportive environment where they felt heard and understood. One participant remarked, “*we usually interact with our clients in that manner, but I’ve never felt that I should be treated the same way. I am touched*”. The participants found sharing experiences with fellow social workers provided relief, fostering a sense of togetherness and belongingness and reducing isolation. Many participants also reported feeling re-energised and motivated after engaging in the programme’s mindfulness practices and self-reflection. A few participants observed that the programme created space for emotional processing and self-care, which aided them to regain a sense of agency and resilience, alleviating some of the exhaustion linked to burnout. One participant expressed that the coaches’ guidance allowed for “*reflecting more deeply on ourselves*” and a better understanding of their own emotions.

Practical techniques such as establishing healthy boundaries, the 3A model (i.e., acknowledge, act, appreciate), and mindfulness exercises, were well received. The participants noted that these techniques encouraged self-discovery, increased self-awareness, and enabled them to recognise the early signs of burnout, paving the road to develop healthier strategies to manage stress and enhance their mental wellness. Furthermore, the programme content was largely seen as relevant and applicable by the participants. The 3A technique, in particular, was deemed helpful in both personal and professional settings as one participant, initially hesitant, shared their application of the practices at work, finding them “*surprisingly helpful*” as they witnessed clients’ “*immediate relief*”, and explained, “*you can sense their (clients’) transition from potentially anxious state to quickly calm down and feel more comfortable at ease*”. More than half of the respondents agreed that the ability to set clearer psychological boundaries is a crucial skill in managing burnout. One participant noted “*(learning) to refrain ourselves from becoming overly invested in clients’ issues, and instead focusing understanding their emotions (and sinking into it)… is quite a rewarding skill to learn*”.

All the participants highlighted the value of the one-on-one coaching in providing them a dedicated space to express emotions and to feel supported. One participant explained that, “*it felt odd at first… that someone would listen to me expressing myself. But having this space and someone genuinely interested in listening to what you have to say is an enjoyable process*”. This non-judgemental, supportive presence helped the participants experience a sense of emotional release and self-reflection. However, some participants found certain sessions challenging, as they brought up deep-seated emotions from past experiences. While one participant preferred less distressing alternatives, most viewed this reflective process as essential for personal growth. One participant recognised that it is “*a lifelong journey*” reflecting on personal experiences, which may resonate with clients’ own issues, and that “*it’s a process (of learning)*”.

Several participants found the daily mindfulness exercises delivered via the WhatsApp group to be helpful reminders for sustaining the programme’s positive effects. Several participants suggested that these prompts made it easier to maintain mindfulness practices without needing to develop their own routines, helping participants manage stress and stay present at work. One participant shared, “*I often have to hurry in the mornings to arrive at work on time, and I listen to it when commuting on public transport*”, which helped them maintain a sense of calmness. A few participants also mentioned the availability of the additional resources offered by the programme organiser, such as YouTube and blogs, providing extra materials for participants to learn and engage in their ongoing practices.

#### 3.2.3. Recommendations for Improving the Programme’s Future Iterations

The participants expressed positive views about the Burn-Not-Out programme. They also highlighted key areas for continued focus and identified aspects for enhancement.

In terms of areas that are worth keeping and continuing to focus on, the participants appreciated the programme’s blended learning approach, which balanced online and in-person sessions, offering flexibility that accommodated their busy schedules while still maintaining the benefits of in-person interactions. One participant noted, “*if all eight sessions were face-to-face, I wouldn’t have been able to participate*. One-on-one coaching was frequently mentioned as essential as it provided a safe space for emotional processing and skills application, which participants found crucial for mitigating burnout. In-class exercises, such as the 3A model and boundary setting practices, were valued for their relevance and practicality, especially the small group activities that reinforced skills development. The participants also underscored the importance of social workers receiving support from their peers and having a platform to share their experiences. This ongoing support, along with daily mindfulness exercises promoting knowledge exchange, can equip them to cope with stress and maintain their long-term well-being. One participant shared, “*we need a space where we can express our emotions freely and safely and build a network where we support fellow social workers*”.

The participants suggested several areas for enhancement. One of the recurring messages was the importance of strengthening peer support networks and fostering a sense of community within the social work profession. The participants suggested establishing a “*community of practice*”, such as regular peer support meetings or a mentorship programme where social workers could continue training to reinforce the skills learned during the programme. One participant emphasised that “*practicing more often after learning these techniques… and receiving feedback on what we could improve (will be helpful)*”. Additionally, the participants recommended incorporating problem-solving training and intergenerational skill-sharing within the profession. One participant noted, “*the younger generation has better creativity than those from my generation… but they may not have our skills in observing and attending clients… Why don’t we foster an environment for intergenerational learning within the social work profession?*”. These training initiatives could better prepare social workers for the realities of the profession, mitigating the risk of burnout.

The participants also provided additional considerations that went beyond the scope of this evaluation. These included integrating training on self-care practices, mindfulness, and boundary setting into the social work curriculum and addressing the systematic issues contributing to social worker burnout, such as streamlining administrative processes, reviewing bureaucratic hurdles, advocating for better compensation, and promoting supportive leadership styles.

## 4. Discussion

This study examined the ProQOL scale and depressive symptoms among social workers in Hong Kong, alongside the initial impact of the Burn-Not-Out programme aimed at reducing burnout within this profession. The results indicated significant improvements in social workers’ mental health outcomes following the programme, supporting our first hypothesis. Specifically, the levels of burnout, STS, and depressive symptoms decreased post-programme. At baseline, 90.4% of the participants reported moderate levels of burnout, which reduced to 73.4% post-programme (t = 7.04, *p* < 0.001). Similarly, the presence of moderate STS reduced from 77.7% to 64.9% after the programme (t = 3.99, *p* < 0.001), and the proportion of participants with no depressive symptoms increased from 36.2% pre-programme to 47.9% post-programme (t = −5.529, *p* < 0.001). The pre-programme results align with global research highlighting the alarming presence of burnout and STS among social workers and other helping professionals globally [12,24,26,39,40,41,42,43,44]. Consistent with previous reviews, our findings reinforce the effectiveness of targeted interventions in mitigating these issues [45,46,47].

Regarding CS, the proportion of participants reporting high levels of CS increased from 5.3% to 14.9% post-programme (t = −5.34, *p* < 0.001). Notably, the participants also demonstrated improvements in boundary setting abilities. The participants’ ability to reject unreasonable requests at work increased from 21.7% to 51.3% (t = −5.71, *p* < 0.001), the ability to differentiate work responsibilities rose from 32.4% to 69.9% (t = −6.59, *p* < 0.001), and the proportion of participants feeling a reduced emotional burden from clients increased from 34.0% to 68.1% (t = −5.02, *p* < 0.001). These results align with our second hypothesis, showing an increased CS and the development of healthier boundaries post-programme. Since a key emphasis of the Burn-Not-Out programme was to help participants shift their perspective on setting healthy boundaries rather than becoming emotionally entangled in service users’ problems, the participants were encouraged to recognise the importance of maintaining professional distance and acknowledging clients’ struggles without internalising their emotions [20]. This approach is vital in preventing compassion fatigue and ensuring long-term emotional resilience in high-stress environments.

We hypothesized that burnout levels would be positively associated with depressive symptoms. Not only was this confirmed (r = 0.66, *p* < 0.001) but we also found significant positive relationships between burnout and all three items related to healthy boundaries (H = 18.07, 18.20, and 10.90; *p* < 0.001). Our findings echo the existing literature, underscoring the critical role of CS in promoting social worker’s well-being and work–life balance [48] and the importance of setting boundaries when providing mental health services [49].

The qualitative findings support the existing literature that highlights the negative impact of job stress on social workers’ personal lives, job satisfaction, and physical and mental health [50]. It appeared that the participants relied on their intrinsic motivations to find satisfaction by helping those in need, highlighting the importance of CS in the field of social work [48,51]. While the evidence has suggested mindfulness-based interventions could reduce stress levels and build resilience [52] and cognitive-behavioural interventions and relaxation techniques have been effective in alleviating emotional exhaustion [53], some studies have argued that they only provided short-term relief [54] as barriers such as high workloads and insufficient managerial support can limit the effectiveness of these interventions [55].

Fostering a sense of community among social workers emerged as another crucial factor in improving their mental well-being. Evidence supports the notion that organisational interventions promoting community and camaraderie can enhance professional fulfilment and reduce burnout [56]. During the COVID-19 pandemic, maintaining professional connections was essential in supporting social workers through unprecedented challenges [57]. Therefore, creating a safe space where social workers can express emotions, receive peer support, and engage in mutual learning is vital for preventing burnout and ensuring the profession’s long-term sustainability.

However, two key limitations should be noted. The quantitative analysis was restricted to 94 sets of paired data, as many participants did not provide matching IDs for the pre- and post-surveys. This led to discarding 178 completed surveys, which reduced this study’s sample size and potentially affected the generalisability of the results. Second, the focus group recruitment process adopted a cohort sampling method, which may have disproportionately attracted participants with a positive predisposition towards the programme, limiting the diversity of perspectives captured in the qualitative findings.

In the given context of the increasing demands and challenges faced by social workers both globally and locally in Hong Kong, maintaining the mental health of social workers is increasingly critical. This study contributes to the limited existing literature on this topic by providing baseline data on the ProQOL scale and depressive symptoms among Hong Kong social workers, evaluating the effectiveness of the Burn-Not-Out programme, and exploring strategies to safeguard their well-being. Effective interventions to alleviate burnout and depressive symptoms are essential to address the complex challenges facing this vital workforce. Given the selfless nature of helping professionals, who often prioritise the needs of others above their own, we strongly recommend that Hong Kong social workers prioritise self-care and incorporate mindfulness practices to enhance self-efficacy, fostering a compassionate and healthy working environment.

This study aimed to advocate for and raise awareness of the presence of occupational burnout among social workers in Hong Kong and to call for more empirical strategies to reconcile the needs of service providers and users. Our findings highlight the critical need for a systemic approach to enhance social workers’ well-being. Integrating structured mental health support and continuous professional development programmes focused on self-care within social work organisations will be instrumental in mitigating burnout. Furthermore, policies advocating for manageable caseloads, adequate staffing, and equitable compensation are vital for retaining qualified professionals and sustaining the social work workforce. Future research should involve larger scale studies that cover a boarder range of social workers beyond programme participants and investigate the role of psychological capital in fostering hope, resilience, and well-being, which have been shown to buffer against job stress and burnout [23,58]. By identifying the signs, addressing root causes, setting healthy boundaries, and seeking support, social workers can improve their quality of life both professionally and personally and, in turn, enhance the quality of care they provide to the communities they serve.

## Figures and Tables

**Table 1 behavsci-14-00915-t001:** Participants’ demographics and job-related backgrounds.

		*n* (%)		Mean (S.D.)	Range
Total		94 (100.0)	Age	41.11 (10.00)	23.0–63.0
			Years of work	14.96 (9.56)	0.4–40.0
			Average work hours	44.16 (10.66)	0.0–65.0
Gender					
	Male	13 (13.8)	Age	41.77 (9.84)	30.0–61.0
			Years of work	13.92 (8.09)	5.0–32.0
			Average work hours	44.04 (5.04)	30.0–50.0
	Female	79 (84.0)	Age	41.10 (10.16)	23.0–63.0
			Years of work	15.16 (9.86)	0.4–40.0
			Average work hours	41.12 (11.47)	0.0–65.0
Occupation					
	Social worker	88 (93.6)			
	Counsellor	3 (3.2)			
	Other	3 (3.2)			
Scope of service					
	Children and youth services	28 (29.8)			
	Elderly services	21 (22.3)			
	Community services	17 (18.1)			
	Family services	10 (10.6)			
	Rehabilitation services	6 (6.4)			
	Others	12 (12.8)			

Note: *n*—valid paired sample.

**Table 2 behavsci-14-00915-t002:** Correlations between dependent and independent variables.

		Age	Years of Work	Pre-Programme				Post-Programme			
				Burnout	STS	CS	Depress	Burnout	STS	CS	Depress
Age		1									
Years of work		0.888 **	1								
Pre	Burnout	−0.138	−0.211 *	1							
	STS	0.118	0.087	0.624 **	1						
	CS	0.223 **	0.259 **	−0.647 **	−0.199 **	1					
	Depress	−0.56	−0.57	0.661**	0.461 **	−0.359 **	1				
Post	Burnout	−0.005	−0.087	0.537 **	0.248 *	−0.348 **	0.529 **	1			
	STS	0.064	0.103	0.301 **	0.485 **	0.001	0.292 **	0.475 **	1		
	CS	0.270	0.164	−0.518 **	−0.169	0.638 **	−0.367 **	−0.701 **	0.067	1	
	Depress	−0.043	−0.016	0.377 **	0.218 *	−0.174	0.623 **	0.640 **	0.440 **	−0.332 **	1

Note: STS—secondary traumatic stress; CS—compassion satisfaction; Depress—depressive symptoms; **—correlation is significant at the 0.01 level (two-tailed); and *—correlation is significant at the 0.05 level (two-tailed).

**Table 3 behavsci-14-00915-t003:** Pre-programme Kruskal–Wallis H tests, pairwise comparisons, and effect size.

		Kruskal–Wallis H Test	Disagree–Unsure		Unsure–Agree		Disagree–Agree	
			Pairwise (*p*-Value)	Effect Size (r)	Pairwise (*p*-Value)	Effect Size (r)	Pairwise (*p*-Value)	Effect Size (r)
Pre-programme								
Burnout								
	Boundary 1	18.073 **	0.7990	0.1319	0.003	0.4175	0.000	0.5540
	Boundary 2	18.199 **	1.0000	0.0168	0.000	0.4693	0.003	0.4496
	Boundary 3	10.904 **	0.0440	0.3070	1.000	0.1112	0.004	0.4207
STS								
	Boundary 1	17.042 **	0.0800	0.2630	0.076	0.2862	0.000	0.5500
	Boundary 2	10.401 **	0.9970	0.1221	0.042	0.2899	0.008	0.4147
	Boundary 3	13.514 **	0.0260	0.3299	0.001	0.1398	0.758	0.4724
CS								
	Boundary 1	7.307 *	0.9250	0.1209	0.194	0.2365	0.021	0.3594
	Boundary 2	6.869 *	1.0000	0.1115	0.161	0.2275	0.039	0.3417
	Boundary 3	3.688	0.5910	0.1625	1.000	0.0855	0.173	0.2493
Depress								
	Boundary 1	16.309 **	0.8340	0.1288	0.006	0.3942	0.000	0.5271
	Boundary 2	35.609 **	1.0000	0.0224	0.000	0.6398	0.000	0.6598
	Boundary 3	18.834 **	0.0220	0.3373	0.186	0.2280	0.000	0.5681
Post-programme								
Burnout								
	Boundary 1	25.198 **	1.0000	0.0050	0.000	0.4984	0.001	0.4401
	Boundary 2	11.784 **	1.0000	0.0952	0.003	0.3531	0.513	0.1613
	Boundary 3	5.146	0.5177	0.2411	0.865	0.1145	0.092	0.2584
STS								
	Boundary 1	6.085 *	0.3266	0.2417	0.049	0.2706	1.000	0.0216
	Boundary 2	5.752	0.2751	0.3187	0.076	0.2386	1.000	0.0624
	Boundary 3	5.589	1.0000	0.1407	0.280	0.1809	0.158	0.2317
CS								
	Boundary 1	13.885 **	0.6130	0.1914	0.036	0.2827	0.002	0.4172
	Boundary 2	7.158 *	1.0000	0.0007	0.043	0.2614	0.474	0.1664
	Boundary 3	0.92	1.0000	0.0904	1.000	0.0558	1.000	0.1060
Depress								
	Boundary 1	13.591 **	1.0000	0.1290	0.001	0.4020	0.169	0.2367
	Boundary 2	4.571	0.2981	0.3114	0.169	0.2035	1.000	0.0799
	Boundary 3	1.311	1.0000	0.0130	0.947	0.1082	1.000	0.0863

Note: STS—secondary traumatic stress; CS—compassion satisfaction; Depress—depressive symptoms; **—Kruskal–Wallis H test is significant at the 0.01 level (two-tailed); *—Kruskal–Wallis H test is significant at the 0.05 level (two-tailed); Boundary 1—“I can refuse unreasonable work request”; Boundary 2—“I can distinguish work that is of my responsibilities from those that is not my responsibilities”; and Boundary 3—“I won’t carry the emotions and responsibilities of my clients”.

**Table 4 behavsci-14-00915-t004:** Paired changes in outcome variables before and after the programme and effect size.

						Paired Differences				
		Low (%)	Moderate (%)		High (%)	Mean (S.D.)	*t*-Test	C.I. (Lower–Upper)		Effect Size (d)
Burnout	Pre	9.6	90.4		0.0	3.1596 (4.3535)	7.036 **	2.27–4.05		0.726
	Post	26.6	73.4		0.0					
STS	Pre	22.3	77.7		0.0	1.9468 (4.7255)	3.994 **	0.98–2.91		0.412
	Post	35.1	64.9		0.0					
CS	Pre	2.1	92.6		5.3	−0.1444 (0.2528)	−5.536 **	−0.20–−0.09		0.571
	Post	1.1	84.0		14.9					
Boundary 1	Pre	35.1	10.4		24.5	−0.4787 (0.8129)	−5.710 **	−0.65–−0.31		0.589
	Post	16.0	30.8		53.2					
Boundary 2	Pre	23.4	43.6		33.0	−0.5426 (0.7987)	−6.586 **	−0.71–−0.38		0.679
	Post	6.4	23.4		70.2					
Boundary 3	Pre	28.7	38.3		33.0	−0.4787 (0.9243)	−5.021 **	−0.67–−0.29		0.518
	Post	8.5	25.5		66.0					
							**Paired Differences**			
		**None**	**Mild**	**Moderate**	**Mod–Severe**	**Severe**	**Mean (S.D.)**	***t*-Test**	**C.I.** **(Lower–Upper)**	**Effect** **Size (d)**
Depress	Pre	36.2	34.0	24.5	4.3	1.1	−0.144 (0.2526)	−5.529 **	−0.2–−0.09	0.570
	Post	47.9	39.4	11.7	1.1	0.0				

Note: STS—secondary traumatic stress; CS—compassion satisfaction; Depress—depressive symptoms; Boundary 1—“I can refuse unreasonable work request”; Boundary 2—“I can distinguish work that is of my responsibilities from those that is not my responsibilities”; Boundary 3—“I won’t carry the emotions and responsibilities of my clients”; **—paired-*t* test is significant at the 0.01 level (two-tailed).

**Table 5 behavsci-14-00915-t005:** Focus group discussion—participants’ characteristics.

		No. of Participants
Gender		
	Male	2
	Female	9
Years of work		
	Less than 1 year	1
	7 years	1
	8 years	1
	12 years	1
	20 years	2
	25 years or more	2
	Did not disclose	3
Scope of service		
	Elderly	3
	School-based	3
	Children and youth	2
	Community	1
	Family	1
	Rehabilitation	1

**Table 6 behavsci-14-00915-t006:** Content analysis—themes and subthemes.

No. of Codes			No. of Codes		
Theme 1: Burnout			Theme 6: Course mental wellness		
	Fulfilling organisational requirements	19		Being cared for	6
	The caregiver role and client dependency	5		Re-energised	5
Theme 2: Secondary traumatic stress				Self-discovery and reflection	4
	Emotional exhaustion	7		Witnessing colleagues getting emotional	2
	Concern for clients’ safety	3	Theme 7: Skillset and application		
Theme 3: Interpersonal relations				In-class exercise relevance	7
	Offering support	11		Setting boundaries	5
	Managing seniors and subordinates	7		One-on-one coaching	4
	Lack of support	2	Theme 8: Sustainability		
Theme 4: Other occupational stress				Daily mindfulness reminder	4
	Socio-political environment	5		Availability of additional resources	2
	Self-perceived incompetence	4			
Theme 5: Compassion satisfaction					
	Meaningful tasks are worthwhile	7			
	Short-term satisfaction	2			

## Data Availability

The data that support the findings of this study are available from the corresponding author upon reasonable request.

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
