# Peer review of "The Evaluation of a Mindful Coaching Programme to Reduce Burnout in Social Workers in Hong Kong—A Pilot Study"

_behavsci, 2024, doi:10.3390/bs14100915_

Round 1
Reviewer 1 Report
Comments and Suggestions for Authors
As stated by the authors in their abstract, the "study examined the prevalence of burnout, secondary traumatic stress, and depressive symptoms among social workers in Hong Kong and evaluated the effectiveness of the Burn-Not-Out programme in reducing their level of burnout." This topic is highly relevant, and the research findings presented in the article could have implications beyond the national borders of the country where the study was conducted. The article is generally clear and well-structured in its methodological choices and arguments, and its content aligns with the journal's aims. I have a couple of notes and suggestions for revision:
1. It is not clear what the scope and implementation framework of the Burn-Not-Out programme are. While Dream Possible is mentioned as the program's creator, the nature of this organization is not clearly defined.
2. Provide a more detailed explanation of the methods used to analyze the outcomes of the semi-structured focus group discussions.
Reviewer 2 Report
Comments and Suggestions for Authors
I have included my feedback in the attached word document.

Comments on the Quality of English LanguageThere are minor spelling and grammatical errors.
Reviewer 3 Report
Comments and Suggestions for Authors
The paper needs English language proofreading.
Abstract: In general needs to be revised to reflect the study clearly and concisely.
· Please add the name of the tools.
· Add statistical values of significance level in the result section
· Delete limitations from abstract
1.1. The programme: Please involve this section with materials and methods.
Materials and Methods: In general, needs to be reorganized and concise. There is so much information. Please revise to be sure of accurate months for data collection.
· Lines 87-90: The study hypothesis needs to be added to the introduction.
· Sitting and participants: who are the participants, from were recruited? What are the inclusion and exclusion criteria?
· Add the reference for Healthy boundaries and the Cronbach alpha, please.
· Sample size calculation is missing.
· What is the methods of qualitative data analysis?
Result: In general, some parts need to be transferred with methodology.
· I suggest deleting Figure 1. Please merge Tables 1 and 2 to be presented more clearly.
· Lines 183-184 are not clear
· I suggest revising the statistics and presentation of the figures.
· The qualitative results need to be presented in tables showing for example, themes, sub-themes, and coding.
· Add a table with characteristics of qualitative data participants, please.
· Qualitative results is lengthy please revise to be concise.
Discussion: Please revise after revising the result section. Please reflect the accurate result and match it with abstract and result sections. Discussion needs to provide support for current study and adding disagreeing studies with rationales.
References: some old references need to be updated to be less than 10 years.
Comments on the Quality of English LanguageThe paper needs English language proofreading.
Round 2
Reviewer 2 Report
Comments and Suggestions for Authors
The authors have adequately addressed the points raised in the first round of review.
Reviewer 3 Report
Comments and Suggestions for Authors
Dear Authors,
Thank you for taking the comments positivly which enhance the quality of the paper. The paper is satisfactory for publication.
Thanks